# Piezoelectricity and topological quantum phase transitions in two-dimensional spin-orbit coupled crystals with time-reversal symmetry

Jiabin Yu[1] & Chao-Xing Liu[1 ✉]

Finding new physical responses that signal topological quantum phase transitions is of both theoretical and experimental importance. Here, we demonstrate that the piezoelectric response can change discontinuously across a topological quantum phase transition in two-dimensional time-reversal invariant systems with spin-orbit coupling, thus serving as a direct probe of the transition. We study all gap closing cases for all 7 plane groups that allow non-vanishing piezoelectricity, and find that any gap closing with 1 fine-tuning parameter between two gapped states changes either the $Z_2$ invariant or the locally stable valley Chern number. The jump of the piezoelectric response is found to exist for all these transitions, and we propose the HgTe/CdTe quantum well and $BaMnSb_2$ as two potential experimental platforms. Our work provides a general theoretical framework to classify topological quantum phase transitions, and reveals their ubiquitous relation to the piezoelectric response.

[1] Department of Physics, the Pennsylvania State University, University Park, PA 16802, USA. ✉email: cxl56@psu.edu

The discovery of topological phases and topological phase transitions has revolutionized our understanding of quantum states of matter and quantum phase transitions[1–3]. Two topologically distinct gapped phases cannot be adiabatically connected; if the system continuously evolves from one phase to the other, a topological quantum phase transition (TQPT) with the energy gap closing (GC) must occur. A direct way to probe such TQPTs is to detect the discontinuous change of certain physical response functions. Celebrated examples include the jump of the Hall conductance across the plateau transition in the integer quantum Hall system[4,5], the jump of the two-terminal conductance across the TQPT between the quantum spin Hall (QSH) state and normal insulator (NI) state in a two-dimensional (2D) time-reversal (TR) invariant system[6], and the jump of the magnetoelectric coefficient across the TQPT between the strong topological insulator phase and NI phase in a 3D TR invariant system[7–10]. The physical responses in all these examples are induced by the electromagnetic field. A natural question then arises: can we detect TQPTs with other types of perturbation?

Here we theoretically answer this question in the affirmative: the discontinuous change of the piezoelectric response is a ubiquitous and direct signature of 2D TQPTs. The piezoelectric effect, the electric charge response induced by the applied strain, is characterized by the piezoelectric tensor (PET) to the leading order. PET was originally defined to relate the change of the charge polarization $P$ with the infinitesimal homogeneous strain, which reads[11]

$$\gamma_{ijk} = \frac{\partial P_i}{\partial u_{jk}}\bigg|_{u_{jk}\to 0}, \tag{1}$$

where $u_{ij} = (\partial_{x_i} u_j + \partial_{x_j} u_i)/2$ is the strain tensor and $u_i$ is the displacement at $\mathbf{x}$. The modern theory of polarization[12–14] later identified the above definition as improper[15] due to the ambiguity of $\mathbf{P}$ in crystals, while the proper definition adds the adiabatic time dependence to $u_{jk}$ and relates it to the bulk current density $J_i$ that can change the surface charge:

$$\gamma_{ijk} = \frac{\partial J_i}{\partial \dot{u}_{jk}}\bigg|_{u_{jk}, \dot{u}_{jk}\to 0}. \tag{2}$$

With Eq. (2), the PET of an 2D insulating crystal has been derived as[15,16]

$$\gamma_{ijk} = -e \int \frac{d^2 k}{(2\pi)^2} \sum_n F^n_{k_i, u_{jk}}\bigg|_{u_{jk}\to 0}, \tag{3}$$

where the integral is over the entire first Brillouin zone (1BZ), and $n$ ranges over all occupied bands. The $F^n_{k_i, u_{jk}}$ term has a Berry-curvature-like expression

$$F^n_{k_i, u_{jk}} = (-i)\Big[\langle \partial_{k_i}\varphi_{n,\mathbf{k}}|\partial_{u_{jk}}\varphi_{n,\mathbf{k}}\rangle - (k_i \leftrightarrow u_{jk})\Big] \tag{4}$$

with $|\varphi_{n,\mathbf{k}}\rangle$ the periodic part of the Bloch state in the presence of the strain. (See the Methods for more details.) The expression indicates an extreme similarity between Eq. (3) and the expression for the Chern number (CN)[5]. It is this similarity that motivates us to study the relation between the PET and the TQPT.

Despite the similarity, the topology connected to the PET is essentially different from the CN, since the PET can exist in TR invariant systems whose CNs always vanish. We, in this work, study the piezoelectric response of 2D TR invariant systems in the presence of the significant spin-orbit coupling (SOC) and demonstrate the jump of all symmetry-allowed PET components across the

TQPT. In particular, we focus on the 7 out of the 17 plane groups (PGs) that allow non-vanishing PET components[17,18], including $p1$, $p1m1$, $c1m1$, $p1g1$, $p3$, $p3m1$, and $p31m$. The two-fold rotation $C_2$ (with the axis perpendicular to the 2D plane) or the 2D inversion restricts the PET to zero in the other 10 PGs[19], according to $\gamma_{ijk} = \sum_{i'j'k'} R_{ii'} R_{jj'} R_{kk'} \gamma_{i'j'k'}$ for any $O(2)$ symmetry $R$ of the 2D material. Through a systematic study, we find that any GC between two gapped states that only requires 1 fine-tuning parameter is a TQPT in the sense that it changes either the $Z_2$ index[1,2] or the valley CN[20]. Although the change of the valley CN is locally stable[21], we still treat the corresponding GC as a TQPT, since the two states cannot be adiabatically connected when the valley is well defined. All the TQPTs contain no stable gapless phase in between two gapped phases, and thereby we refer to them as the direct TQPTs. All PET components that are allowed by the crystalline symmetry exhibit discontinuous changes across any of the direct TQPTs, showing the ubiquitous connection. Interestingly, when the gap closes at momenta that are not TR invariant, the strain tensor $u_{ij}$ acts as a pseudo-gauge field[22] at the TQPT, making the PET jump directly proportional to the change of the $Z_2$ index or the valley CN.

Our work presents a general framework for the PET jump across the TQPT in 2D TR invariant systems with SOC. The relation between the PET and the valley CN in the low-energy effective model has been studied in graphene with a staggered potential[23], h-BN[24,25], and monolayer transition metal dichalcogenides (TMDs) $XY_2$ for X = Mo/W and Y = S/Se[25]. However, these early works have not pointed out that it is the PET jump (well described within the low-energy effective model) that is the experimental signature directly related to the TQPT, while the PET itself at fixed parameters might contain the non-topological background given by high-energy bands. Moreover, these works, unlike our systematic study, only considered one specific plane group ($p3m1$) around one specific type of momenta ($K$, $K'$). The relation between the PET and the $Z_2$ index were not explored either. Besides, graphene and h-BN have neglectable SOC, and the TMDs have a large gap, making them not suitable for realizing TQPT. We thereby propose two realistic material systems, the HgTe/CdTe quantum well (QW) and the layered material $BaMnSb_2$, as potential experimental platforms. The $Z_2$ TQPT and PET jump can be achieved by varying the thickness or the gate voltages in the HgTe/CdTe QW or by tuning lattice distortion in $BaMnSb_2$.

## Results

**PET jump across a direct QSH-NI TQPT.** We start from a simple example of the TQPT discussed in ref. [26]. They (in the example of our interest) considered the case with no crystalline symmetries other than the lattice translation (PG $p1$) and focused on the GC at two momenta $\pm\mathbf{k}_0$ that are not TR invariant momenta (TRIM), as labeled by red crosses in Fig. 1a. The low-energy effective theory for the electron around $\mathbf{k}_0$ can be described by the Hamiltonian of a 2D massive Dirac fermion[26]

$$h_{+,0}(\mathbf{q}) = E_0(\mathbf{q})\sigma_0 + v_x q_1 \sigma_x + v_y q_2 \sigma_y + m\sigma_z, \tag{5}$$

where $\mathbf{q} = \mathbf{k} - \mathbf{k}_0$, $m$ is the tuning parameter for the TQPT, and $\sigma$'s are Pauli matrices. In the above Hamiltonian, the unitary transformation on the bases and the scaling/rotation of $\mathbf{q}$ are performed for the simplicity of the Hamiltonian; the latter generally makes $q_1$, $q_2$ along two non-orthogonal directions. (See Supplementary Note 3A for details.) The effective Hamiltonian at $-\mathbf{k}_0$ is related to $h_{+,0}$ by the TR symmetry. After choosing appropriate bases at $-\mathbf{k}_0$, the TR symmetry can

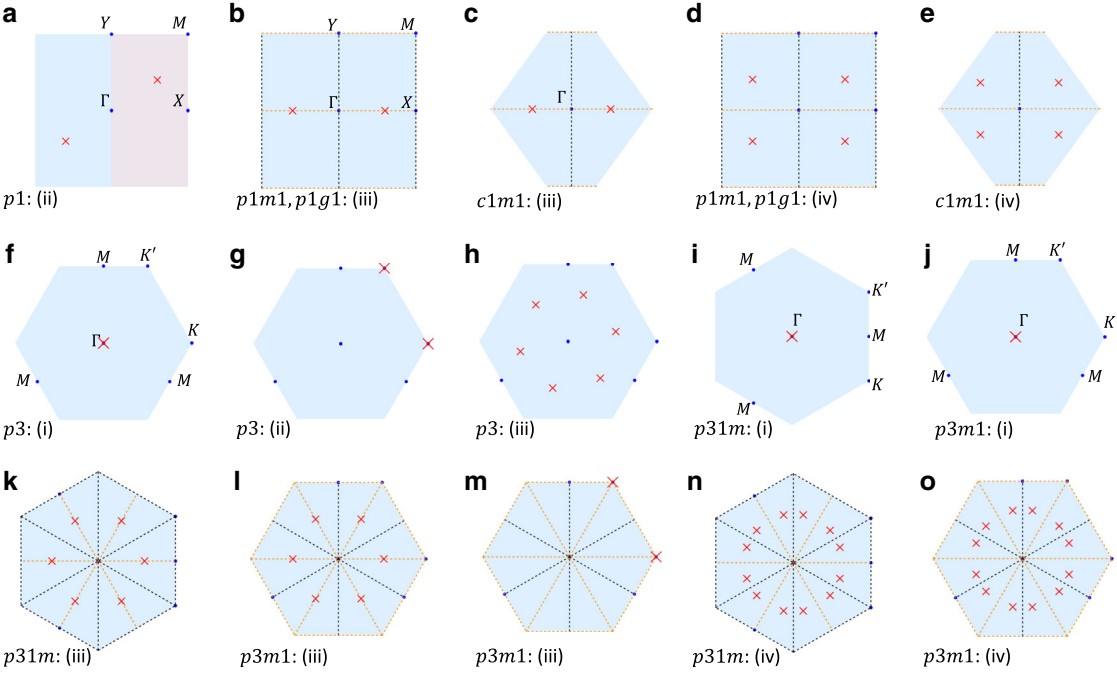

**Fig. 1 GC cases with 1 fine-tuning parameter.** The figure shows the GC cases between insulating states with 1 fine-tuning parameter for all 7 PGs with non-vanishing PET. The red cross labels the GC momenta, the light blue background indicates the 1BZ, and the light red part in **a** indicates the half 1BZ. The black and orange dashed lines label the momenta invariant under the mirror/glide symmetry and the combination of mirror/glide and TR symmetries, respectively. The figures are first grouped according to the PGs and then ordered based on the GC scenarios listed in Table 1, whose labels are next to the names of the PGs.

be represented as $\mathcal{T} \doteq i\sigma_y \mathcal{K}$ with $\mathcal{K}$ the complex conjugate, leading to

$$h_{-,0}(\mathbf{q}) = E_0(-\mathbf{q})\sigma_0 + v_x q_1 \sigma_x + v_y q_2 \sigma_y - m\sigma_z. \tag{6}$$

According to ref. [26], the TQPT between the QSH insulator and the NI (distinguished by the $Z_2$ index) occurs when the mass $m$ in $h_{\pm,0}(\mathbf{q})$ changes its sign. The argument used to determine change of the $Z_2$ index was presented in ref. [27] and is discussed below for integrity. Since there is no inversion symmetry in PG $p1$, the $Z_2$ index can be determined from the CN of the contracted half first Brillouin zone (1BZ), where the half 1BZ is chosen such that its Kramers' partner covers the other half. Specifically, the $Z_2$ index is changed (unchanged) by the GC if the CN of the contracted half 1BZ changes by an odd (even) integer. Without loss of generality, let us choose the half 1BZ to contain $\mathbf{k}_0$, as shown in Fig. 1a. Since $h_{+,0}$ is a 2D gapped Dirac Hamiltonian, the CN of the contracted half 1BZ changes by $\Delta N_+ = -\text{sgn}(v_x v_y)$ as $m$ increases from $0^-$ to $0^+$, featuring a direct QSH-NI TQPT as $v_x v_y$ is typically nonzero.

We next discuss the piezoelectric effect in this simple effective model. To do so, we need to introduce the electron-strain coupling around $\pm\mathbf{k}_0$ based on the TR symmetry:

$$h_{\pm,1}(u) = \xi_{0,ij}\sigma_0 u_{ij} \pm \xi_{a',ij}\sigma_{a'} u_{ij}, \tag{7}$$

where the duplicated indexes, including $a' = x, y, z$ and $i, j = 1, 2$, are summed over henceforth unless specified otherwise. $\xi$s are the material-dependent coupling constants between the low-energy electrons and the strain tensor, which obey $\xi_{a,ij} = \xi_{a,ji}$ with $a = 0, x, y, z$ owing to $u_{ij} = u_{ji}$ and are related to the electron-phonon coupling[28]. The full form of the effective Hamiltonian is

then given by

$$h_\pm(\mathbf{q}, u) = h_{\pm,0}(\mathbf{q}) + h_{\pm,1}(u). \tag{8}$$

To use Eq. (3), we simplify Eq. (8) by neglecting the $E_0$ term, which has no influence on the piezoelectric response of insulators (see Supplementary Note 1). When $\xi_{x,ij} = \xi_{y,ij} = 0$, the Hamiltonian $h_\pm$ has effective inversion symmetry within each valley, $\sigma_z h_\pm(-\mathbf{q}, u)\sigma_z = h_\pm(\mathbf{q}, u)$, which forbids the piezoelectric effect. Thus, $\xi_{0,ij}$ and $\xi_{z,ij}$ terms cannot contribute to the PET, and neglecting them leads to a further simplified version of Eq. (8):

$$h_\pm(\mathbf{q}, u) = \left[ v_x(q_1 \pm A_1^{pse}) \right]\sigma_x + \left[ v_y(q_2 \pm A_2^{pse}) \right]\sigma_y \\ \pm m\sigma_z, \tag{9}$$

where $A_1^{pse} = \xi_{x,ij} u_{ij}/v_x$ and $A_2^{pse} = \xi_{y,ij} u_{ij}/v_y$. The above form suggests that the remaining strain terms, $\xi_{x,ij}$ and $\xi_{y,ij}$, serve as the pseudo-gauge field $A_i^{pse}$ that has opposite signs for two valleys $\pm\mathbf{k}_0$[10,22,25,29]. As the strain tensor only exists in the form of $q_i \pm A_i^{pse}$, the derivative with respect to $u_{ij}$ in Eq. (3) can be replaced by the derivative with respect to the momentum as

$$\partial_{u_{ij}} |\varphi_{\pm,\mathbf{q}}\rangle = \frac{\partial A_{i'}^{pse}}{\partial u_{ij}} \partial_{A_{i'}^{pse}} |\varphi_{\pm,\mathbf{q}}\rangle = \pm \frac{\partial A_{i'}^{pse}}{\partial u_{ij}} \partial_{q_{i'}} |\varphi_{\pm,\mathbf{q}}\rangle, \tag{10}$$

where $\varphi_\pm$ are the occupied bands of $h_\pm$. Substituting the above equation into Eq. (3) leads to

$$\gamma_{1ij}^{eff} = -e \int \frac{d^2 q}{(2\pi)^2} \sum_{\alpha = \pm} \alpha F_{12}^\alpha(\mathbf{q}) \frac{\partial A_2^{pse}}{\partial u_{ij}} \tag{11}$$

$$\gamma_{2ij}^{eff} = e \int \frac{d^2 q}{(2\pi)^2} \sum_{\alpha = \pm} \alpha F_{12}^\alpha(\mathbf{q}) \frac{\partial A_1^{pse}}{\partial u_{ij}},$$

where $F_{12}^{\pm}(\mathbf{q})$ is the conventional Berry curvature of the occupied band of $h_{\pm}(\mathbf{q}, 0)$. The superscript *eff* means that we neglect the contribution from bands beyond the effective model Eq. (8), indicating that the above equation is not the complete PET. Nevertheless, it can accurately give the PET change across the TQPT since high-energy bands experience an adiabatic deformation and the corresponding background PET contribution should remain unchanged at the transition ($m = 0$). As $m$ varies from $0^-$ to $0^+$, Eq. (11) gives the change of PET $\Delta\gamma_{ijk}$ as

$$\Delta\gamma_{1ij} = -e\frac{\Delta N_+}{\pi}\frac{\xi_{y,ij}}{v_y}$$
$$\Delta\gamma_{2ij} = e\frac{\Delta N_+}{\pi}\frac{\xi_{x,ij}}{v_x}. \tag{12}$$

The PET jump shown in the above equation is nonzero since $v_x v_y$ and the electron-strain coupling $\xi$s are typically non-zero. We thus conclude that for $p1$ group, a jump of PET that is directly proportional to the change of the $Z_2$ index occurs across the TQPT, when the gap closes not at TRIM.

The PET jump can be physically understood based on Eq. (2). Let first focus on one GC momentum, say $\mathbf{k}_0$. Since the strain tensor couples to the electron in the way similar to the $U(1)$ gauge field as shown in Eq. (8), $\dot{u}_{jk}$ should act like a electric field on the electron. According to Eq. (2), $\gamma_{ijk}$ should then behave like the Hall conductance, whose jump is proportional to the change of CN $\Delta N_+$. Now we include the other GC momentum $-\mathbf{k}_0$. Unlike the actual $U(1)$ gauge field, the pseudo-gauge field given by the strain couples oppositely to the electron at the two GC momenta (Eq. (8)). The opposite signs of the coupling can cancel the opposite signs of the Berry curvature, and thus, in contrast to the actual Hall conductance, the contributions to $\gamma_{ijk}$ from $\pm\mathbf{k}_0$ add up to a nonzero value instead of canceling each other, leading to the non-zero topological jump in Eq. (12).

**Classification of direct 2D TQPTs and PET jumps for 7 PGs.** The above section discusses an example of 2D QSH-NI TQPT for the $p1$ PG and illustrates the main picture of the relation between the 2D TQPT and the PET jump. It is well-known that the crystalline symmetry imposes strong constraints on the PET[19] (see the Methods). Topological states in different space/plane groups have been classified based on the topological quantum chemistry[30–36], the symmetry indicator[37–40], and other early methods[41–43]. On the contrary, only a small number of works[26,38,44,45] have studied the crystal symmetry constraint on the GC forms of the TQPTs. While the GC between non-degenerate states was studied in ref. [45] for various layer groups in the presence of TR symmetry and SOC, the GC that involves degenerate states, like between two Kramers' pairs, has not been explored. In particular, the topology change and the PET jump across any GC case with codimension 1 (or equivalently requiring 1 fine-tuning parameter) have not been discussed. As the

substrate, on which the 2D materials are grown, typically reduces layer groups to PGs by breaking the extra symmetries, a study based on PG is typically enough for experimental predictions. Therefore, we next present a comprehensive study on the GC forms of TQPTs in all 7 PGs that allow nonvanishing PET, namely $p1$, $p1m1$, $c1m1$, $p1g1$, $p3$, $p31m$, and $p3m1$. The main results are summarized in Fig. 1 and Table 1, as discussed below. The other 10 PGs ($p2$, $p2mm$, $p2mg$, $p2gg$, $c2mm$, $p4$, $p4mm$, $p4gm$, $p6$, and $p6mm$) have vanishing PET due to the existence of inversion symmetry or $C_2$ rotation symmetry, and are briefly discussed in Supplementary Note 2E.

TQPTs in different PGs can be analyzed in the following three steps. In the first step, we classify the GC based on the GC momenta and the symmetry property of the bands involved in the GC. To do so, we define the group $\mathcal{G}_0$ for a GC momentum $\mathbf{k}_0$ such that $\mathcal{G}_0$ contains all symmetry operations that leave $\mathbf{k}_0$ invariant (including the little group of $\mathbf{k}_0$ and TR-related operations). We start with a coarse classification based on $\mathcal{G}_0$, which leads to 2 scenarios for $p1$, 3 scenarios for $p3$, and 4 scenarios for $p1m1$, $c1m1$, $p1g1$, $p31m$, and $p3m1$, as listed in Table 1 and the Methods. To illustrate this classification, we consider the $p3$ group as an example, which contains 3 different scenarios. In scenario (i), the GC is located at TRIM ($\mathcal{T} \in \mathcal{G}_0$), i.e., the $\Gamma$ point or three $M$ points in Fig. 1f. In scenario (ii), the GC occurs simultaneously at $K$ and $K'$ where $\mathcal{G}_0$ contains $C_3$ but no $\mathcal{T}$ (Fig. 1g). In scenario (iii), the GC occurs at six generic momenta ($\mathcal{G}_0$ only contains lattice translations) that are related by $C_3$ rotation and TR (Fig. 1h). The classification of GC momenta is coarse here since $\mathcal{G}_0$ can still vary within one scenario. For example, in scenario (i) of $p3$, $\mathcal{G}_0$ at $\Gamma$ contains $C_3$ while $\mathcal{G}_0$ at $M$ does not. Moreover, even at a certain GC momentum with a certain $\mathcal{G}_0$, the symmetry properties of bands involved in the GC may vary. For example, at $K$ in scenario (ii) of $p3$, the gap may close between two states with the same or different $C_3$ eigenvalues. Therefore, we further refine our classification by taking these subtleties into consideration and classify each GC scenario into finer GC cases.

In the second step, for each GC case, we construct a symmetry-allowed low-energy effective Hamiltonian that well captures the GC and count the number of fine-tuning parameters. Since $\mathcal{G}_0$ and the symmetry properties of the bands involved in the GC are fixed in one GC case, the form of the effective Hamiltonian can be unambiguously determined (see details in Supplementary Notes 2 and 3). After obtaining the effective Hamiltonian, we can count the number of fine-tuning parameters required for each GC and select out all GC cases that require only 1 fine-tuning parameter (or equivalently has codimension 1), as shown in Fig. 1. Only these cases can be direct TQPTs between two gapped phases, since any two gapped states in the parameter space are adiabatically connected if 2 or more fine-tuning parameters are required to close the gap, and 0 codimension means there is a stable gapless phase in between two gapped phases. Our analysis shows that all GC cases in scenarios (i) for $p1$, (i) and (ii) for $p1m1$, $c1m1$, and $p1g1$, and (ii) for $p3m1$ and $p31m$ need 0

**Table 1 Summary for all 7 PGs with non-vanishing PET.**

| PGs | $p1$ | | $p1m1$, $c1m1$, $p1g1$ | | | | $p3$ | | | $p3m1$, $p31m$ | | | |
|---|---|---|---|---|---|---|---|---|---|---|---|---|---|
| Scenario | (i) | (ii) | (i) | (ii) | (iii) | (iv) | (i) | (ii) | (iii) | (i) | (ii) | (iii) | (iv) |
| Codim-1 GC | × | (a) | × | × | (b and c) | (d and e) | (f) | (g) | (h) | (i and j) | × | (k–m) | (n and o) |
| Topo. Inv. | N/A | $Z_2$ | N/A | N/A | $Z_2$ | VCN | $Z_2$ | $Z_2$ | $Z_2$ | $Z_2$ | N/A | $Z_2$ | VCN |
| PET Jump | N/A | ✓ | N/A | N/A | ✓ | ✓ | ✓ | ✓ | ✓ | ✓ | N/A | ✓ | ✓ |

The scenarios are classified by the symmetries that leave the GC momenta invariant, as shown in the Methods. Codim-1 GC means the GC cases with 1 fine-tuning parameter or codimension 1. If at least one GC case between gapped states with 1 fine-tuning parameter exists in the corresponding scenario, the subfigures in Fig. 1 that illustrate the GC momenta are referred to; otherwise, we fill in a ×.
Topo. Inv. labels the topological invariant changed by the codimensoin-1 GC, $Z_2$ means the $Z_2$ index, and VCN means the corresponding case changes the valley CN when the valley is well-defined.

fine-tuning parameter or more than 1 fine-tuning parameters and thus cannot correspond to the direct TQPTs, while codimension-1 GC cases can exist in all other scenarios.

In the third and final step, we demonstrate the topological nature of all the codimension-1 GC cases by evaluating the change of certain topological invariants and derive the corresponding PET jump. As shown in Table 1, the $Z_2$ index is changed in all codimension-1 GC cases of scenarios (ii) for $p1$, (iii) for $p1m1$, $c1m1$, and $p1g1$, (i)-(iii) for $p3$, and (i) and (iii) for $p3m1$ and $p31m$, while the valley CN is changed for all codimension-1 GC cases of the scenarios (iv) for $p1m1$, $c1m1$, $p1g1$, $p3m1$, and $p31m$. We would like to emphasize that although valley CN itself is in general not quantized in a gapped phase, the change of valley CN across a gap closing is quantized and has physical consequence[46]. (See the "Methods" section for more details.) According to Fig. 1, the $Z_2$ cases either close the gap at TRIM or have an odd number of Dirac cones in half 1BZ, while all the valley CN cases (Fig. 1d, e and Fig. 1n, o) have an even number of Dirac cones in half 1BZ, forbidding the change of the $Z_2$ index. Nevertheless, no matter which type, they all lead to discontinuous changes of the symmetry-allowed PET components (see detailed calculation of PET in Supplementary Note 2).

In sum, we conclude that for all 7 PGs with non-vanishing PET, all the GC cases between two gapped phases with 1 fine-tuning parameter are direct TQPTs that change either $Z_2$ index or valley CN, and they all induce the discontinuous change of the symmetry-allowed PET components. Based on these results, we propose the following criteria to find realistic systems to test our theoretical predictions: (i) whether it breaks the 2D inversion or two-fold rotation with axis perpendicular to the 2D plane, (ii) whether it has significant SOC, and (iii) whether there is a tunable way to realize the GC. Applying these conditions to the existing material systems for 2D TQPTs, we find two realistic material systems, namely the HgTe/CdTe QW and the layered material BaMnSb$_2$, which are studied in the following.

**HgTe/CdTe QW.** It has been demonstrated[6,47] that the TQPT between the QSH insulator and NI phases in the HgTe/CdTe QW can be achieved by tuning the HgTe thickness $d$. Tuning applied electric field $\mathcal{E}$ was theoretically predicted as an alternative way to achieve TQPT[48,49], making the system an ideal platform to study the PET jump at TQPTs. Here, the stacking direction of the QW is chosen to be (111) instead of the well-studied (001) direction[50], since the latter would allow a two-fold rotation that forbids PET. Without the applied electric field, the (111) QW has the TR symmetry and the $C_{3v}$ symmetries (generated by three-fold rotation along (111) and the mirror perpendicular to $(\bar{1}10)$); adding electric field along (111) does not change the symmetry properties. We should then expect one independent symmetry-allowed PET component $\gamma_{222}$ similar to Eq. (18) in the Methods, where 2 labels the direction $(11\bar{2})$.

The electronic band structure of the (111) QW can be described by the 6-band Kane model with the bases $(|\Gamma_6, \pm\frac{1}{2}\rangle, |\Gamma_8, \pm\frac{3}{2}\rangle, |\Gamma_8, \pm\frac{1}{2}\rangle)$. The electric field $\mathcal{E}$ along (111) can be introduced by adding a linear electric potential that is independent of orbitals and spins. In this electron Hamiltonian, there are two inversion-breaking (IB) effects, the inherent IB effect in the Kane model and the applied electric field, and we neglect the former for simplicity. Note that such approximation does not lead to vanishing PET even for $\mathcal{E} = 0$ because the IB electron-strain coupling will be kept.

We first discuss the inversion-invariant $\mathcal{E} = 0$ case and focus on the PET jump induced by varying the width $d$. In this case, there are two doubly degenerate bands closest to the Fermi energy, namely $|E_1, \pm\rangle$ and $|H_1, \pm\rangle$ bands with opposite parities. With the method proposed in ref. [6], we find that the gap between two bands closes at the $\Gamma$ point around $d = 65\text{Å}$ as shown in Fig. 2a. The GC must be a $Z_2$ TQPT owing to the opposite parities of the two bands, and it belongs to scenario (i) of $p3m1/p31m$ discussed in Table 1 and the Methods. We further include the electron-strain coupling, and numerically plot the independent

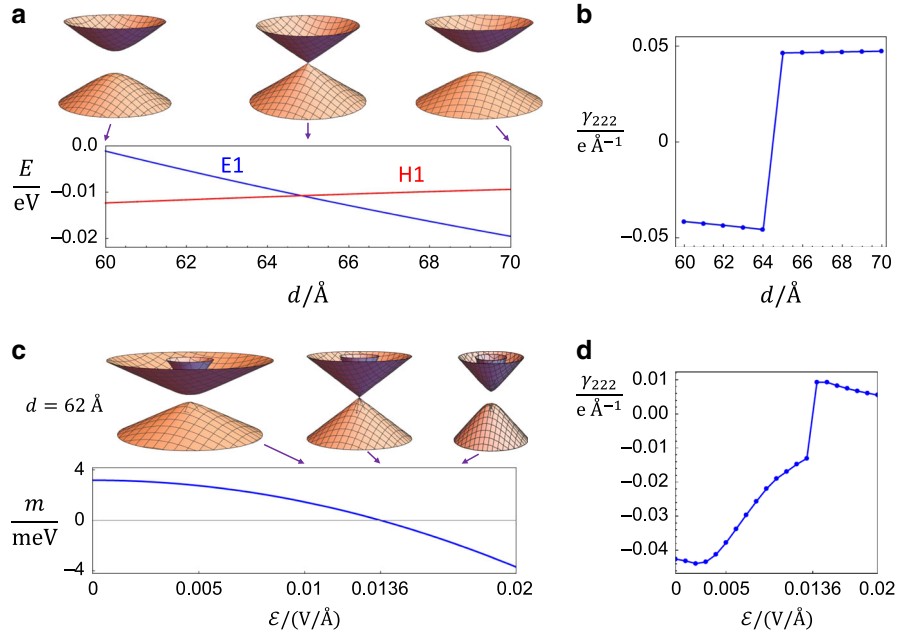

**Fig. 2 HgTe/CdTe QW.** This figure shows the energy dispersion and the PET of the HgTe QW with the stacking direction (111). In **a**, the lower panel shows the energy of E1 (blue) and H1 (red) bands at $\Gamma$ point as a function of the width $d$, and the upper panel shows the energy dispersion at $d = 60, 65, 70$ Å from left to right, respectively. The GC happens around $d \approx 65$ Å, which is slightly different form the well-known $d = 63$ Å reported in ref. [47] for the (001) stacking direction owing to the anisotropy effect. **b** The PET component $\gamma_{222}$ as a function of $d$. In **c**, the lower panel plots the gap $m$ as a function of the electric field $\mathcal{E}$ with $d = 62$ Å, showing that the gap closes at $\mathcal{E} \approx 0.0136$ V Å$^{-1}$. The upper panel of **c** demonstrates the energy dispersion at $\mathcal{E} = 0.01, 0.0136, 0.017$ V Å$^{-1}$ from left to right, respectively. **d** The PET component $\gamma_{222}$ as a function of $\mathcal{E}$.

PET component $\gamma_{222}$ as the function of the width in Fig. 2b, which shows a jump around $d = 65$ Å (see Supplementary Note 5).

Next we study the TQPT induced by the applied electric field. In order to realize the GC at a nonzero value of the electric field, we fix the width of the QW at $d = 62$ Å, away from 65 Å. After adding the linear electric potential along (111) in the 6-band Kane model, we numerically find that the GC at $\Gamma$ point happens at $\mathcal{E} \approx 0.0136$ V Å$^{-1}$, as shown in Fig. 2c. Such GC belongs to scenario (i) of $p3m1/p31m$ and is still a $Z_2$ TQPT since the extra IB term cannot influence the $Z_2$ topology change. The PET component $\gamma_{222}$ is numerically shown in Fig. 2d, showing the jump across the TQPT. The PET jump in Fig. 2b, d has the order $10 \sim 100$ pC m$^{-1}$, and thus is possible to be probed by the current experimental technique[51].

**Layered material BaMnSb₂.** BaMnSb₂ is a 3D layered material that consists of Ba-Sb layers and Mn-Sb layers, which are stacked alternatively along the (001) direction (or equivalently $z$ direction). The electrons in $p_x$ and $p_y$ orbitals of Sb atoms in the Ba-Sb layers account for the transport of the material. Owing to the insulating Mn-Sb layers, the tunneling along the $z$ direction among different Ba-Sb layers is much weaker than the in-plane hopping terms, and thus BaMnSb₂ can be treated as a quasi-2D material[52]. Therefore, we can only consider one Ba-Sb layer, whose structure is shown in Fig. 3a. Owing to the zig-zag distortion of the Sb atoms (solid lines in Fig. 3a), the symmetry group that captures the main physics is spanned by the TR symmetry $\mathcal{T}$ and two mirror operations $m_y$ and $m_z$ that are perpendicular to $y$ and $z$ axes, respectively. It turns out that the mirror symmetry $m_z$ does nothing but guarantee the z-component of the spin to be a good quantum number in the low energy[52], allowing us to view the system as a spin-conserved TR-invariant 2D system with PG $p1m1$. Slightly different from the demonstration in the Methods, the mirror here is perpendicular to $y$ instead of $x$, and thereby PG $p1m1$ now requires $\gamma_{yyy} = \gamma_{yxx} = \gamma_{xyx} = \gamma_{xxy} = 0$ and leaves the other four components as symmetry-allowed.

To describe this system, a tight-binding model with $p_x$ and $p_y$ orbitals of Sb atoms was constructed in ref. [52] based on the first-principle calculation, and the form of the model is reviewed in Supplementary Note 6A for integrity. This model qualitatively captures all the main features of the electronic band structure of BaMnSb₂. The key parameter of the model is the distortion parameter $\alpha$ that describes the zig-zag distortion of the Sb atoms. When $\alpha$ is tuned to a critical value $\alpha_c \approx 0.86$, the gap of the system closes at two valleys $\mathbf{K}_\pm = (\pi, \pm k_{y0})$ near $X$ along $X - M$ in the BZ, as shown in Fig. 3b. This GC results in a TQPT between the QSH state and the NI state in one Ba-Sb layer, as confirmed by the direct calculation of $Z_2$ index (Fig. 3c) according to expression in ref. [53]. Since the two GC momenta are invariant under $\mathcal{T}m_y$, this GC case satisfies the definition of scenario (iii) for $p1m1$. We further numerically verify the PET jump induced by the GC with the tight-binding model. The jump of the symmetry-allowed PET components is found at the TQPT around $\alpha = \alpha_c$ in Fig. 3d, while the components forbidden by the symmetry stay zero. According to Fig. 3d, both the jump and background are of the same order of magnitude, 0.1 eÅ$^{-1}$ for $\gamma_{yxy,yyx}$ and 0.01 eÅ$^{-1}$ for $\gamma_{xxx,xyy}$, indicating that the jump is experimentally measurable. The $Z_2$ topology change and the PET jump can also be analytically verified based on the effective model as discussed in Supplementary Note 6.

## Discussion

In conclusion, we demonstrate that for all PGs that allow non-vanishing PET, the piezoelectric response has a discontinuous change across any TQPT in 2D TR invariant systems with significant SOC. Potential material realizations include the HgTe/CdTe QW and the layered material BaMnSb₂.

The early study on MoS₂ has demonstrated that the values of the PET obtained from the effective model might be (though not always) quite close to those from the first principles calculations[25]. Therefore, although our theory is based on the effective Hamiltonian, the predicted jump of the PET is quite likely to be significant and even the sign change of PET, such as Fig. 2b, d for the HgTe case, might exist in realistic materials. The evaluation of the PET from the first principles calculations is left for the future.

Although we only focus on two realistic material systems in this work, the theory can be directly applied to other material systems. For example, the calculations for the HgTe/CdTe QW are also applicable to InAs/GaSb QWs, which share the same model[54]. The QSH effect has also been observed in the monolayer 1T'-WTe₂[55-57], but its inversion symmetry[58] forbids the piezoelectric effect. Therefore, a significant inversion breaking effect from the environment (such as substrate) is required to test our prediction in this system. While the SOC strength in graphene is small, it has been shown that the bilayer graphene sandwiched by TMDs has enhanced SOC and serves as a platform to observe TQPT[59,60], where the PET jump is likely to exist. The piezoelectric effect has been observed in several 2D material systems[51,61,62], and therefore, the material systems and the experimental technique for the observation of the PET jump are both available. Since the PET jump is directly related to the TQPT, it further provides a new experimental approach to extract the critical exponents and universality behaviors of the TQPT, which can only be analyzed through transport measurements nowadays.

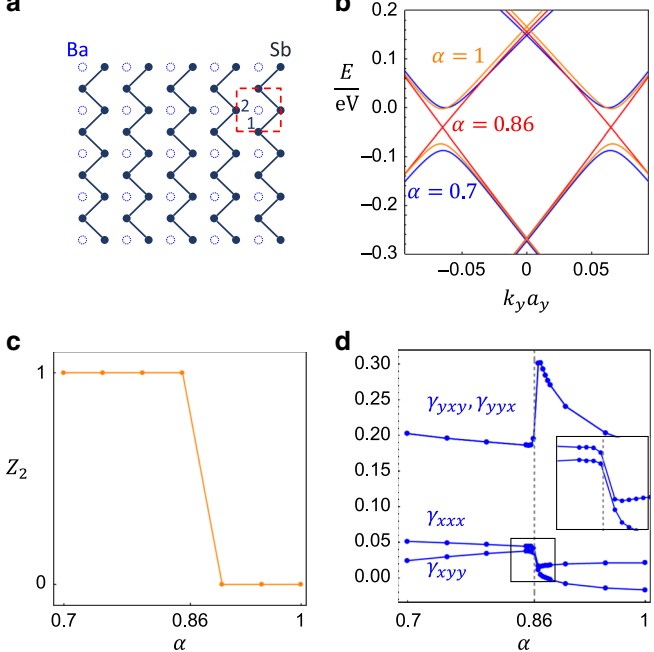

**Fig. 3 Layered Material BaMnSb₂. a** Illustration of the Ba-Sb layer, where each dashed circle stands for the projection of two Ba atoms onto the Sb layer and the solid dots are Sb atoms. The solid lines connecting Sb atoms indicate the zig-zag distortion, and the red dashed box marks the unit cell with 1 and 2 labeling the two Sb atoms. **b** The band structure of the TB model for BaMnSb₂ along $M - X - M$ for $\alpha = 0.86$ (red), $\alpha = 1$ (orange), and $\alpha = 0.7$ (blue), respectively, where $X$ is at $k_y = 0$. In **c** and **d**, the $Z_2$ index and PET components obtained from the TB model are plotted as a function of $\alpha$, respectively. In **d**, the PET components are in the unit eÅ$^{-1}$, the gray dashed line is at $\alpha = 0.86$, and the inset is the zoom-in version of the boxed region.

This work only focuses on 2D TR invariant systems with SOC, and the generalization to systems without SOC, without TR symmetry, or in 3D is left for the future. Despite the similarity between Eq. (3) and the expression of CN, the generalization to TR-breaking systems with non-zero CNs requires caution, due to the change of the definition of polarization[63]. Another interesting question is whether the PET jump exists across the transition between states of different higher-order[64–67] or fragile topology[34,68]. We notice that although the dynamical piezoelectric effect may exist in metallic systems[69], its description is different from Eq. (3). It is thus intriguing to ask how the dynamical PET behaves across the transitions between insulating and semimetal phases.

## Methods

**Expression for the PET**. According to refs. [15,63], the expression for the PET of insulators, Eq. (3), is derived for systems with zero CNs and within the clamped-ion approximation where ions exactly follow the homogeneous deformation and thus cannot contribute to the PET. Even though the ion contribution might be non-zero in reality, the approximation is still legitimate in our study of PET jump since the ion contribution varies continuously across the GC of electronic bands.

Eq. (3) involves the derivative of the periodic part of the Bloch state $|\varphi_{n,\mathbf{k}}\rangle$ with respect to the strain tensor $u_{jk}$. $|\varphi_{n,\mathbf{k}}\rangle$ can always be expressed as $|\varphi_{n,\mathbf{k}}\rangle = \sum_{\mathbf{G}} f_{n,\mathbf{k},\mathbf{G}}|\mathbf{G}\rangle$ with $\mathbf{G}$ the reciprocal lattice vector, and the derivative in fact means $|\partial_{u_{jk}}\varphi_{n,\mathbf{k}}\rangle \equiv \sum_{\mathbf{G}}(\partial_{u_{jk}}f_{n,\mathbf{k},\mathbf{G}})|\mathbf{G}\rangle$[15]. In this way, the ill-defined $\partial_{u_{ij}}|\mathbf{G}\rangle$ is avoided, despite that $|\mathbf{G}\rangle$ is not continuous as changing the strain. If replacing the $|\partial_{u_{jk}}\varphi_{n,\mathbf{k}}\rangle$ in Eq. (3) by a momentum derivative $|\partial_{k_j}\varphi_{n,\mathbf{k}}\rangle$ with $j$ different from $i$, the PET expression transforms into $-eC\epsilon^{ij}/(2\pi)$, where $\epsilon^{ij} = -\epsilon^{ji}$, $\epsilon^{xy} = 1$, and $C$ is the Chern number of the 2D insulator[5]

$$C = \int \frac{d^2k}{2\pi} \sum_n F^n_{k_x,k_y}. \tag{13}$$

This reveals the similarity between the PET expression and the expression of the CN.

**PG $p1$**. For $p1$, no special constraints are imposed on the PET. There are two GC scenarios for the PG $p1$ with TR symmetry:

(i) gap closes at TRIM ($\mathcal{T} \in \mathcal{G}_0$),
(ii) gap closes not at TRIM ($\mathcal{T} \notin \mathcal{G}_0$).

In scenario (ii), $\mathcal{G}_0$ contains no symmetries other than the lattice translation, which we refer to as the trivial $\mathcal{G}_0$.

**PGs $p1m1$, $c1m1$, and $p1g1$**. All three PGs, $p1m1$, $c1m1$, and $p1g1$, are generated by a mirror-related symmetry $\mathcal{U}$ and the lattice translation. $\mathcal{U}$ is a mirror operation for $p1m1/c1m1$ and a glide operation for $p1g1$. The difference between $p1m1$ and $c1m1$ lies on the directions of the primitive lattice vectors relative to the mirror line, which is not important for our discussion here. Without loss of generality, we choose the mirror or glide line to be perpendicular to $x$, labeled as $m_x$ or $g_x$, respectively. The glide operation is thus denoted as $g_x = \{m_x|0\frac{1}{2}\}$, where $0\frac{1}{2}$ represents the translation by half the primitive lattice vector along $y$. The $\mathcal{U}$ symmetry in these three PGs requires

$$\gamma_{ijk} = (-1)^i(-1)^j(-1)^k\gamma_{ijk} \tag{14}$$

with $(-1)^x = -1$ and $(-1)^y = 1$, resulting that $\gamma_{xxx} = \gamma_{xyy} = \gamma_{yxy} = \gamma_{yyx} = 0$ while $\gamma_{xxy}$, $\gamma_{xyx}$, $\gamma_{yxx}$, $\gamma_{yyy}$ are allowed to be nonzero. For the symmetry analysis here, the PET behaves the same under the glide and mirror operations since $u_{ij}$ is considered in the continuum limit. Based on $\mathcal{G}_0$, we obtain in total 4 GC scenarios for these three PGs:

(i) the GC at TRIM ($\mathcal{G}_0$ contains $\mathcal{T}$),
(ii) $\mathcal{G}_0$ contains $\mathcal{U}$ but not $\mathcal{T}$,
(iii) $\mathcal{G}_0$ contains $\mathcal{U}\mathcal{T}$ but not $\mathcal{T}$,
(iv) $\mathcal{G}_0$ is trivial.

**PG $p3$**. PG $p3$ is generated by 3-fold rotation $C_3$ and the lattice translation. Owing to $C_3$, the PET satisfies the following relation

$$\gamma_{ijk} = \sum_{i'j'k'} [R(C_3)]_{ii'}[R(C_3)]_{jj'}[R(C_3)]_{kk'}\gamma_{i'j'k'}, \tag{15}$$

where

$$R(C_3) = \begin{pmatrix} -\frac{1}{2} & -\frac{\sqrt{3}}{2} \\ \frac{\sqrt{3}}{2} & -\frac{1}{2} \end{pmatrix}. \tag{16}$$

Solving the above equation gives two independent components $\gamma_{xxx}$ and $\gamma_{yyy}$ as

$$\gamma_{yxy} = \gamma_{yyx} = \gamma_{xyy} = -\gamma_{xxx} \tag{17}$$

$$\gamma_{xxy} = \gamma_{xyx} = \gamma_{yxx} = -\gamma_{yyy}.$$

Again, we classify the GC for $p3$ according to $\mathcal{G}_0$, resulting in three different scenarios:

(i) $\mathcal{G}_0$ contains $\mathcal{T}$,
(ii) $\mathcal{G}_0$ contains $C_3$ but not $\mathcal{T}$,
(iii) $\mathcal{G}_0$ is trivial.

Here we do not have a scenario for $\mathcal{G}_0$ containing $C_3\mathcal{T}$ but no $\mathcal{T}$, since $(C_3\mathcal{T})^3$ is equivalent to $\mathcal{T}$.

**PGs $p31m$ and $p3m1$**. Both PGs $p31m$ and $p3m1$ are generated by the lattice translation, the three-fold rotation $C_3$, and a mirror symmetry which we choose to be $m_x$ without loss of generality. The difference between the two PGs lies on the direction of the mirror line relative to the primitive lattice vector: the mirror line is parallel or perpendicular to one primitive lattice vector for $p31m$ or $p3m1$, respectively. $C_3$ and $m_x$ span the point group $C_{3v}$, which makes the PET satisfy Eq. (14) and Eq. (15). As a result, we have

$$\gamma_{xxx} = \gamma_{xyy} = \gamma_{yxy} = \gamma_{yyx} = 0 \tag{18}$$

$$\gamma_{xyx} = \gamma_{xxy} = \gamma_{yxx} = -\gamma_{yyy}$$

for the PET, and thus $\gamma_{yyy}$ serves as the only independent symmetry-allowed PET component. We classify the GC scenarios into 4 types according to $\mathcal{G}_0$:

(i) $\mathcal{G}_0$ contains $\mathcal{T}$,
(ii) $\mathcal{G}_0$ contains at least one of the three mirror symmetry operations in $C_{3v}$ (again labeled as $\mathcal{U} = m_x$, $C_3m_x$, or $C_3^2m_x$) but no $\mathcal{T}$,
(iii) $\mathcal{G}_0$ contains $\mathcal{U}\mathcal{T}$ but no $\mathcal{T}$,
(iv) $\mathcal{G}_0$ is trivial.

**Valley CN**. In all the valley CN cases (Fig. 1d, e, n, o), the GC points are at generic positions in the 1BZ. The valleys can be physically defined as the positions where the Berry curvature diverges as the gap approaches to zero. The positions of the Berry curvature peaks around the gap closing can be clearly seen in numerical calculations, as long as those peaks are well separated in the momentum space. (See Supplementary Note 4 for more details.) With the positions of the valleys determined, the valley CN on one side of the GC is not necessarily quantized to integers since the integral of Berry curvature is not over a closed manifold. However, the change of valley CN across the GC is always integer-valued, since it is equal to the CN of the Hamiltonian given by patching the two low-energy effective models on the two sides of the GC at large momenta, which lives on a closed manifold. One physical consequence of the quantized change of valley CN is the gapless domain-wall mode[46], which can be experimentally tested with transport or optical measurements[70]. We verify the quantized change of valley CN and demonstrate the corresponding gapless domain-wall mode with a tight-binding model in Supplementary Note 4.

The above argument relies on the constraint that the valleys are well separated in 1BZ, preventing the two states from being adiabatically connected. Without the contraint of well-defined valleys, the valleys are allowed to be merged, and two phases with different valley CNs might be adiabatically connected. Therefore, we refer to the topology characterized by valley CN as locally stable[21], though globally unstable. Nevertheless, we restrict all valleys to be well-defined in our discussion and refer to the corresponding gap closing case as a TQPT.

## Data availability
The datasets generated during and/or analyzed during the current study are available from the authors on reasonable request.

## Code availability
The mathematica code generated during and/or analyzed for the current study are available from the authors on reasonable request.

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

## Acknowledgements

We are thankful for the helpful discussion with B. Andrei Bernevig, Xi Dai, F. Duncan M. Haldane, Shao-Kai Jian, Biao Lian, Xin Liu, Laurens W. Molenkamp, Zhiqiang Mao, Xiao-Qi Sun, David Vanderbilt, Jing Wang, Binghai Yan, Junyi Zhang, and Michael Zaletel. We acknowledge the support of the Office of Naval Research (Grant No. N00014-18-1-2793), the U.S. Department of Energy (Grant No. DESC0019064) and Kaufman New Initiative research grant KA2018-98553 of the Pittsburgh Foundation.

## Author contributions

C.X.L. conceived the original idea and supervised the whole project. J.Y. refined the idea and performed the theoretical analysis, as well as the model calculations. Both authors participated in the manuscript preparation.

## Competing interests

The authors declare no competing interests.
