## [Peer Review File · Nature Communications]

Reviewers' comments:

Reviewer #1 (Remarks to the Author):

Reviewer Comments for NCOMMS-19-38988

"Piezoelectricity and Topological Quantum Phase Transitions in Two-Dimensional Spin-Orbit Coupled Crystals with Time-Reversal Symmetry"

This manuscript details a very novel and general approach to detecting topological quantum phase transitions, specifically by investigating the piezoelectric response of different materials and/or combinations of materials. This systematic study is the first kind that I've come across and the differences between the present study and previous studies are clearly spelled out in the final paragraph of the introduction. I would recommend publishing after the authors address a few questions and help clarify some points to ultimately appeal to a broader audience with interests in materials modeling, system design, and functional materials chemistry.

Introduction:

1) Add a reference (or two?) in the introduction for the 17 plane groups for the general readers of Nature Communications, beyond the 1974 work of R. L. E. Schwarzenberger, because this article will be of interest to scientists in fields beyond solid state physics.

2) In the introduction you mention that Eq. 3 is similar to the expression for Chern number. This is a nice connection to make, but it would be helpful to add the actual expression (Eq. 5 from Ref. 5?) and perhaps a sentence or two describing the similarities. This would be helpful and serve as additional motivation. (Also, as a follow up to this question: In the conclusion, if you revisit the connection between the expression for Chern number and the piezoelectric tensor that you made in the introduction, is there anything else of note to mention given your derivations and results?)

3) Do you have references for the HgTe/CdTe QW and BaMnSb₂ systems that would be helpful starting points, even if it alluded as to why you chose these systems?

HgTe/CdTe Quantum Well subsection:

4) It would be helpful for materials modeling to know a few more details about this system, and this can be brought up in the manuscript or the supplemental materials. Examples would include the crystallographic space groups of the system(s) and how the Hg and Cd are arranged in the QW. Are they layered or in more of a rocksalt-type ordering? The QW is Hg_{0.3}Cd_{0.7}Te | HgTe | Hg_{0.3}Cd_{0.7}Te, but what would happen if you changed composition between $0.8 > \text{Cd} > 0.6$? Would there be a significant change in d for this (111) ordered system?

5) Why fix the width of the QW at 62 Å. if the jump is around 65 Å.? Please discuss this detail, as it's very important to know how the distances are (potentially) related.

Layered Material BaMnSb₂ subsection:

6) BaMnSb₂ is in the SmCuP₂ structure type, which is found in I4/mmm (space group 139) symmetry. Are you implying that because of the zigzag layered structure and a "stuffing" atom that it's ok to consider only the metal-Sb zigzag layered nets in your tight-binding model? The details here are a little confusing and Figure 3a does not help clarify your point. What happens to Mn here, or is it just a spacer for your Ba-Sb network?

7) Could your findings be extrapolated to other materials classes with a similar setup of layers and stuffing atoms between them? For example the PbClF structure type (P4/nmm symmetry, space group 129) contains NaZnSb and NaMnBi, and the BaZn₂P₂ structure type (I4/mmm symmetry, space group 139) contains TiCu₂Se₂, RbCo₂Se₂; would they also be of interest to screen for the piezoelectric jump?

8) Figure 1: How is this figure organized? It's difficult to draw any physical insights in its current form. It might help here to have the scenarios from the supplemental materials in addition to the PG label, and perhaps a sentence or two more explanation as to their ordering?

9) Table 1: You outline multiple PET jump scenarios, but don't explicitly state it in the manuscript in a way that is accessible to materials chemists or crystallographers. Some of these scenarios may already exist in known materials or composites, so any guidance as to where to look would be novel and very helpful. Given your results, do you have any suggestions, crystallographic symmetries, or combinations of elements within a structure that might be fruitful to search?

Reviewer #2 (Remarks to the Author):

The manuscript theoretically proposes a way of detecting topological phase transitions using piezoelectric response. Although there are many theoretical studies about the method for detecting topological phases themselves, there are limited studies on the way of detecting the topological phase transitions. In the manuscript, the authors have considered 2D plane groups including spin-orbit coupling and performed systematic classification of accidental band crossings which can induce nonzero piezoelectric response. This referee thinks that searching physical responses that can detect topological phase transitions is important, and the fact that the piezoelectric response can be generally used for such a purpose is quite interesting. However, this referee has several concerns about the novelty of this paper due to (i) the overlap of the results with a preceding work, (ii) the limitation of the proposed idea in real experiments related with the non-quantized nature of the piezoelectric response. Below this referee gives several questions and comments, which should be properly addressed by the authors, before any decision can be made about the manuscript.

(1) In the section "Classification of Dirac 2D TQPTs and PET jumps for 7 PGs", the authors wrote that "In particular, a systematic classification of GC forms for 2D PGs has not been implemented." However, this is not true. This referee finds one paper [Park and Yang, "Classification of accidental band crossings and emergent semimetals in two-dimensional noncentrosymmetric systems", PRB 96, 125127 (2017)]. As for the classification of band crossing, the work done by Park and Yang presents more general results since all 49 layer groups lacking inversion, not just 17 plane groups, are fully considered. The authors should properly cite this paper, and explain in what sense the reported results are new.

(2) Related with the comment above, depending on the symmetry, an accidental band crossing can sometimes induce a transition from an insulator to a semimetal, not an insulator-insulator transition, which is not considered in the manuscript. What happens in the piezoelectric tensor in these cases?

(3) Considering the numerical results shown in Fig.2 and 3, the piezoelectric tensors are not quantized in general. This means that even in a single gapped phase, piezoelectric tensors can be changed when parameters describing the systems are varied. In this case, how meaningful is the description based on the low energy theory near the gap-closing points? The contributions from the other parts of the Brillouin zone cannot be neglected in general. The authors should give some estimate comparing the contribution from the low energy Hamiltonian and that from the other parts of the band structure, at least for the model considered in the manuscript.

(4) I do not understand how meaningful the valley Chern number is. In particular, considering that the location of the gap-closing points can be anywhere in the Brillouin zone even in a single phase, the physical interpretation of the meaning of the valleys is not clear. In this situation, the corresponding Chern number cannot have a solid meaning.

(5) Symmetry conditions to have nonzero piezoelectric response should be more clearly explained in the main text, especially related with the plane group discussed in the paper.

(6) There are too many acronyms, which disturb the readability of the manuscript.

Reviewer #3 (Remarks to the Author):

In this manuscript, Yu and Liu have thoroughly studied the piezoelectric response across topological phase transitions in two dimensional time-reversal invariant systems. Using both analytical arguments and numerical calculations, the authors have made the strong statement that the piezoelectric response would exhibit a jump across the topological transitions in all 2D time-reversal invariant insulators which break inversion symmetry (C_{2z} symmetry). Moreover, the authors have classified the band-touching events for all 2D plane groups breaking inversion symmetry (preserving time-reversal symmetry), which will be very useful for future theoretical research on 2D materials. The authors also illustrate these ideas using specific material examples, i.e., HgTe/CdTe (111) thin films, and BaMnSb₂, which would provide useful guidelines to experimentalists. The calculations for BaMnSb₂ are indeed quite reliable and practical. Therefore, I would strongly recommend this paper for publication on Nature Communications. I have two questions and suggestions as follows:

1) Does Eq.(3) apply to systems that break time-reversal (TR) symmetry? Or does it rely on the existence of a smooth gauge throughout the Brillouin zone? If it is a general formula which also applies to TR-breaking systems and is completely gauge independent, then it would be interesting to discuss the change of the piezoelectric response across the topological transition in a Chern insulator with broken inversion symmetry.

2) There are both electronic and ionic contributions to the piezoelectric response. The former is what has been discussed in this manuscript. The ionic contribution may be important in a given insulating state, but the change of the ionic contribution to the piezoelectric response is expected to be small and continuous across the topological transition, and this distinguishes the electronic contributions (to the piezoelectric response) as a unique probe of the topological transition. I think it would be nice if the authors could point this out.

Note: All references are cited according to the updated version of the manuscript.

Reply to Reviewer #1

Reviewer #1 (Remarks to the Author):

Reviewer Comments for NCOMMS-19-38988

“Piezoelectricity and Topological Quantum Phase Transitions in Two-Dimensional Spin-Orbit Coupled Crystals with Time-Reversal Symmetry”

This manuscript details a very novel and general approach to detecting topological quantum phase transitions, specifically by investigating the piezoelectric response of different materials and/or combinations of materials. This systematic study is the first kind that I've come across and the differences between the present study and previous studies are clearly spelled out in the final paragraph of the introduction. I would recommend publishing after the authors address a few questions and help clarify some points to ultimately appeal to a broader audience with interests in materials modeling, system design, and functional materials chemistry.

Introduction:

1) Add a reference (or two?) in the introduction for the 17 plane groups for the general readers of Nature Communications, beyond the 1974 work of R. L. E. Schwarzenberger, because this article will be of interest to scientists in fields beyond solid state physics.

Reply: We sincerely appreciate the positive comments of this referee.

As for this point, we thank the referee for introducing the 1974 work of R. L. E. Schwarzenberger to us. We think “International Tables for Crystallography” is a good reference. In volume A, the concepts and notations for space groups (including plane groups) are introduced in part 1 and the table of the plane groups is in part 2. We cited both references in the updated version.

2) In the introduction you mention that Eq. 3 is similar to the expression for Chern number. This is a nice connection to make, but it would be helpful to add the actual expression (Eq. 5 from Ref. 5?) and perhaps a sentence or two describing the similarities. This would be helpful and serve as additional motivation. (Also, as a follow

up to this question: In the conclusion, if you revisit the connection between the expression for Chern number and the piezoelectric tensor that you made in the introduction, is there anything else of note to mention given your derivations and results?)

Reply: We thank the referee for the suggestion. We added the expression of Chern number and discussed its connection to Eq. (3) with more detail in the “Method” section.

In the conclusion section of the updated version, we revisited the connection and gave a note on how to generalize our results to time-reversal-breaking systems with non-zero Chern numbers. We emphasized that this generalization requires caution since the definition of the polarization is changed when Chern number is nonzero, as discussed by Coh and Vanderbilt in Ref. [62]. To our knowledge, whether Eq. (3) can be used for Chern insulators remains an open question.

3) Do you have references for the HgTe/CdTe QW and BaMnSb₂ systems that would be helpful starting points, even if it alluded as to why you chose these systems?

Reply: The criteria that we use to choose systems include (i) whether it breaks the 2D inversion or two-fold rotation with axis perpendicular to the 2D plane, (ii) whether it has significant spin-orbit coupling, and (iii) whether there is a tunable way to realize the gap closing. It turns out both (111) HgTe/CdTe quantum well (QW) and BaMnSb₂ satisfy the conditions. Ref. [6, 46, 48] demonstrated the tunable Z₂ transition in HgTe/CdTe quantum well along (001), and we just need to stack them along (111) to have the right symmetry. Ref. [51] includes the experimental, computational, and theoretical study of BaMnSb₂, suggesting it satisfies the conditions. Therefore, we believe those four references are good starting points. Given the importance of the above criteria in searching for the topological phase transition and piezoelectric coefficient jump, we listed them at the beginning of the material discussion in the updated version.

HgTe/CdTe Quantum Well subsection:

4) It would be helpful for materials modeling to know a few more details about this system, and this can be brought up in the manuscript or the supplemental materials. Examples would include the crystallographic space groups of the system(s) and how the Hg and Cd are arranged in the QW. Are they layered or in more of a rocksalt-type ordering? The QW is Hg_{0.3}Cd_{0.7}Te | HgTe | Hg_{0.3}Cd_{0.7}Te, but what would happen if you changed composition between $0.8 > \text{Cd} > 0.6$? Would there be a significant change in d for this (111) ordered system?

Reply: Both HgTe and CdTe have the standard zinc-blende structure, similar to most II-VI or III-V compound semiconductors. The crystallographic space group of both compounds is $F\bar{4}3m$ (space group No. 216). In the QW, HgTe serves as a well and Hg_{1-x}Cd_xTe serves as the barrier. We use $x=0.7$ simply because this ratio was used in early experimental and theoretical studies (e.g. see Ref. [6,46-49]). As a barrier, we only

require $\text{Hg}_{1-x}\text{Cd}_x\text{Te}$ to be a trivial insulator with a relatively large gap. According to Ref. [15] of the supplementary materials, $\text{Hg}_{1-x}\text{Cd}_x\text{Te}$ becomes a trivial insulator as long as $x > 0.17$. This suggests that one can choose a large range of Cd composition. We added this discussion to the supplementary materials.

5) Why fix the width of the QW at 62 Ang. if the jump is around 65 Ang.? Please discuss this detail, as it's very important to know how the distances are (potentially) related.

Reply: The Z_2 transition in the HgTe/CdTe QW can be induced in two ways. One way is to fix the applied electric field and tune the width. When we fix the applied electric field to zero, the transition and the piezoelectric jump happen when tuning the width to 65 ang.

The other way is to fix the width and tune the applied electric field. Since we want to realize the jump at a non-zero applied electric field, we fix the width at 62 ang, which is away from 65 ang. We find the transition at applied electric field = 0.0136 volts/ang, which is experimentally feasible. We can also fix the QW width at other values and the transition just happens at different values of the applied electric field. Experimentally, the QW thickness cannot be tuned continuously, and thus the electric-field-induced phase transition is desirable. We revised the manuscript to make it clearer.

Layered Material BaMnSb₂ subsection:

6) BaMnSb₂ is in the SmCuP₂ structure type, which is found in I4/mmm (space group 139) symmetry. Are you implying that because of the zigzag layered structure and a "stuffing" atom that it's ok to consider only the metal-Sb zigzag layered nets in your tight-binding model? The details here are a little confusing and Figure 3a does not help clarify your point. What happens to Mn here, or is it just a spacer for your Ba-Sb network?

Reply: The structure of the material can be roughly viewed as stacking Mn-Sb layers and Ba-Sb layers alternatively along z. The layers are extended in the x and y directions. We only need to consider the Ba-Sb layers because the Mn-Sb layers, which separate the Ba-Sb layers, are insulating and thus greatly weaken the interlayer tunneling between different Ba-Sb layers. This has been shown experimentally in Ref. [51]: the ratio of the resistance along z to the resistance along x/y is very large (~670).

Although the ideal BaMnSb₂ has space group I4/mmm, recent experiments, like the scanning transmission electron microscopy measurement in Ref. [51], have demonstrated the zig-zag distortion among Sb atoms in Ba-Sb layers of BaMnSb₂, which reduces the space group to I2mm. This zig-zag distortion is essential for the non-vanishing piezoelectricity. In I4/mmm, there is a two-fold rotation along the axis perpendicular to the Ba-Sb layers (i.e. along z), and thus the piezoelectricity of the layers is restricted to zero. The zig-zag distortion breaks this two-fold rotation symmetry along z and then allows the nonvanishing piezoelectricity of the Ba-Sb layers. We

added more discussion on this material in the main text and added the structure of BaMnSb_2 in the supplementary materials.

7) Could your findings be extrapolated to other materials classes with a similar setup of layers and stuffing atoms between them? For example the PbClF structure type ($P4/nmm$ symmetry, space group 129) contains NaZnSb and NaMnBi , and the BaZn_2P_2 structure type ($I4/mmm$ symmetry, space group 139) contains TlCu_2Se_2 , RbCo_2Se_2 ; would they also be of interest to screen for the piezoelectric jump?

Reply: We think the new material predictions proposed by the referee are very interesting. However, as mentioned in the above, it is crucial to make sure (i) the transport properties of the crystal only involve 2D layers that have very weak interlayer tunneling, (ii) those layers have no 2D inversion symmetry or two-fold rotational symmetry that forbids the piezoelectricity, and (iii) the materials have significant spin-orbit coupling. We notice the mentioned materials have two-fold rotational symmetries (or screw-axis symmetries) in all three directions and thus a certain type of lattice distortion is essential to realize our prediction, just like BaMnSb_2 . An experimentally tunable way to realize the topological phase transition in these compounds is also important.

8) Figure 1: How is this figure organized? It's difficult to draw any physical insights in its current form. It might help here to have the scenarios from the supplemental materials in addition to the PG label, and perhaps a sentence or two more explanation as to their ordering?

Reply: We thank the referee for the suggestion. We added the scenario numbers to Fig. 1. The figures are first grouped according to the plane groups, and within the same set plane groups, they are ordered by gap closing momenta, typically in a high-to-low-symmetry order. We added this discussion in the caption of Fig. 1.

9) Table 1: You outline multiple PET jump scenarios, but don't explicitly state it in the manuscript in a way that is accessible to materials chemists or crystallographers. Some of these scenarios may already exist in known materials or composites, so any guidance as to where to look would be novel and very helpful. Given your results, do you have any suggestions, crystallographic symmetries, or combinations of elements within a structure that might be fruitful to search?

Reply: We thank the referee for the suggestion. As nonvanishing piezoelectricity requires to break the 2D inversion or two-fold rotation with axis perpendicular to the 2D plane, any crystalline structure that satisfies this condition is worth considering. In particular, given the large number of 2D materials with hexagonal lattices, we believe $p3$, $p3m1$, and $p31m$ plane groups might be more promising. Since our theory is done for spin-orbit coupled systems, elements that can bring large spin-orbit coupling are

preferred. As discussed in the above reply, we added a paragraph to describe the criteria for the search of realistic material systems.

Reply to Reviewer #2

Reviewer #2 (Remarks to the Author):

The manuscript theoretically proposes a way of detecting topological phase transitions using piezoelectric response. Although there are many theoretical studies about the method for detecting topological phases themselves, there are limited studies on the way of detecting the topological phase transitions. In the manuscript, the authors have considered 2D plane groups including spin-orbit coupling and performed systematic classification of accidental band crossings which can induce nonzero piezoelectric response. This referee thinks that searching physical responses that can detect topological phase transitions is important, and the fact that the piezoelectric response can be generally used for such a purpose is quite interesting. However, this referee has several concerns about the novelty of this paper due to (i) the overlap of the results with a preceding work, (ii) the limitation of the proposed idea in real experiments related with the non-quantized nature of the piezoelectric response. Below this referee gives several questions and comments, which should be properly addressed by the authors, before any decision can be made about the manuscript.

(1) In the section “Classification of Dirac 2D TQPTs and PET jumps for 7 PGs”, the authors wrote that “In particular, a systematic classification of GC forms for 2D PGs has not been implemented.” However, this is not true. This referee finds one paper [Park and Yang, “Classification of accidental band crossings and emergent semimetals in two-dimensional noncentrosymmetric systems”, PRB 96, 125127 (2017)]. As for the classification of band crossing, the work done by Park and Yang presents more general results since all 49 layer groups lacking inversion, not just 17 plane groups, are fully considered. The authors should properly cite this paper, and explain in what sense the reported results are new.

Reply: We thank the referee for stating that “the fact that the piezoelectric response can be generally used for such a purpose is quite interesting” in the introductory part of the report.

We sincerely appreciate the reference introduced by the referee. The PRB paper indeed is a systematic (though incomplete as discussed below) study of 2D quantum phase transitions for noncentrosymmetric layer groups in the presence of time-reversal symmetry and spin orbit coupling, and has overlap with our classification of gap closing scenarios. Therefore, we revised the statement and properly cited the paper in the updated version.

However, we would like to emphasize that the main goal of our work is to understand the relation between topological quantum phase transitions and piezoelectricity in 2D materials, which has never been considered in the literature. The classification of gap closing cases is a supporting part of our work that serves our main goal. This is completely different from the major purpose of the PRB work, which is the classification itself. Therefore, our central results on piezoelectric jump across the topological quantum phase transitions have NOT been considered in the PRB paper.

In the following, we focus on the classification. We would like to highlight several key results that are presented in the classification part of our work but not in the PRB paper, and argue why we believe the plane group study instead of the layer group study is typically enough for experimental predictions.

- (1) For all plane groups that allow non-vanishing piezoelectricity, our complete classification covers various meaningful gap closing cases that are missing in the PRB paper. The classification presented in the PRB paper only includes the gap closing between two one dimensional irreducible representations (1D irreps) and excludes all other possibilities like the gap closing between two 2D irreps. However, we consider all possible irreps and, in particular, find that the codimension-1 (codim-1) gap closing between two 2D irreps not only exists theoretically (between 2 different Kramers' pairs at the Gamma point for plane groups $p3$, $p3m1$, and $p31m$) but also describes the Z_2 topological quantum phase transitions happening in realistic material systems, like HgTe quantum well. On the other hand, within the limit of 1D irreps, some codim-1 gap closing cases presented in our work are missing in the PRB paper, e.g., the gap closing between two single bands with different C_3 eigenvalues at K and K' points for $p3$ and $p3m1$. Therefore, within plane groups, we present a more comprehensive study on the gap closing scenarios.
- (2) The PRB paper does not study the topology change induced by any codim-1 transition between two insulating states, while we present a careful study for all of them. As a result, the related conclusion of our work, any codim-1 gap closing

between two gapped states changes either Z_2 index or valley Chern number for all plane groups that allow piezoelectricity, is not in the PRB paper.

- (3) We believe the plane group study is typically enough for experimental predictions since the 2D materials are always grown on a substrate which typically breaks the extra symmetries in the layer groups, such as the mirror symmetry with mirror plane lying in the 2D material, the two-fold rotational symmetry that flips the 2D material, and so on. In fact, among the 16 realistic material systems listed in the PRB paper, 15 of them are completely described by the plane groups.

(2) Related with the comment above, depending on the symmetry, an accidental band crossing can sometimes induce a transition from an insulator to a semimetal, not an insulator-insulator transition, which is not considered in the manuscript. What happens in the piezoelectric tensor in these cases?

Reply: The referee claims that the gap closing that includes a transition from an insulator to a semimetal is not considered in our work. However, as described in the “Classification of Direct 2D TQPTs and PET jumps for 7 PGs” section, we study the gap closing momenta, the symmetry property of the bands involved in the gap closing, the symmetry-allowed low-energy effective Hamiltonian, and the number of fine-tuning parameters in all gap closing cases for all plane groups that allow piezoelectricity, including the gap closing that drives a gapped phase to a gapless phase.

Since the static polarization and adiabatic current are not well defined in semimetals, the piezoelectricity of them is not described by Eq. (3), but by a completely different theoretical framework as suggested in Ref. [68]. Therefore, despite the interesting question posed by the referee, we believe it is better to leave the study of the piezoelectricity across the insulator-semimetal gap closing cases for future work, as discussed in the “CONCLUSION AND DISCUSSION” section. We think the original version did not make this point clear enough so we revised the manuscript for clearness.

(3) Considering the numerical results shown in Fig.2 and 3, the piezoelectric tensors are not quantized in general. This means that even in a single gapped phase, piezoelectric tensors can be changed when parameters describing the systems are varied. In this case, how meaningful is the description based on the low energy theory near the gap-closing points? The contributions from the other parts of the Brillouin zone cannot be neglected in general. The authors should give some estimate comparing the contribution from the low energy Hamiltonian and that from the other parts of the band structure, at least for the model considered in the manuscript.

Reply: The referee concerns the validity of the low-energy description for the jump of the piezoelectricity and believes that the contribution from other states beyond the model cannot be neglected in general. We argue that, as described in the last paragraph of the introduction, the topological quantity that should be measured in experiments is NOT the piezoelectricity of a material at a fixed condition (in a single

gapped phase) but the change of piezoelectricity when driving the material across a gap closing or a phase transition. Across a gap closing, the states far away, either in energy or in momentum, from the gap closing point evolve adiabatically owing to the finite gap, and thus their contribution to the change is continuous. Therefore, any discontinuous change or jump, if exists, must come from the states near the gap closing, which are well described within the low-energy effective model.

The remaining question becomes how large the piezoelectric jump well described by the effective model is compared with the continuous background given by high-energy states. Since our study on BaMnSb_2 is based on a tight-binding model, the calculation of piezoelectric coefficients includes both low-energy and high-energy contributions. According to Fig. 3(d), both the jump and background are of the order $0.1 \text{ e}/\text{ang}$ for xyx and yyx components and of order $0.01 \text{ e}/\text{ang}$ for xxx and xxy components in BaMnSb_2 . Therefore, in this realistic model, the jump is of the same order of magnitude as the background and thus is experimentally measurable, indicating that the description of the jump based on the low-energy effective model is meaningful. Although similar discussion was present in the conclusion section, we added a more detailed discussion in the main text.

As discussed in the “CONCLUSION AND DISCUSSION” section, the piezoelectric coefficient in the transition metal dichalcogenides has been calculated with both the low-energy effective model and the first principle calculations (see Ref. [23]). The two methods actually give quite close values, demonstrating that the contribution from the low-energy effective model is dominated in these compounds.

(4) I do not understand how meaningful the valley Chern number is. In particular, considering that the location of the gap-closing points can be anywhere in the Brillouin zone even in a single phase, the physical interpretation of the meaning of the valleys is not clear. In this situation, the corresponding Chern number cannot have a solid meaning.

Reply: The referee concerns about the meaning of the valley Chern number. Although the gap-closing points can locate at generic positions in the first Brillouin zone, the valleys can be physically defined as the positions where the Berry curvature diverges as the gap approaches to zero. The positions of the Berry curvature peaks round the gap closing can be clearly seen in numerical calculations, once different Berry curvature peaks are well separated in the momentum space. With the positions of the valleys determined, the valley Chern number in a single phase is of course not necessarily quantized to integers since the integral of the Berry curvature is not over a closed manifold. However, the change of the valley Chern number is always integer-valued, since it is equal to the Chern number of the Hamiltonian given by patching the two low-energy effective models of two phases at large momenta, which lives on a closed manifold. We believe the above points were not clarified in the previous version and thus we added the relevant discussion to the “Method” section in updated version.

(5) Symmetry conditions to have nonzero piezoelectric response should be more clearly explained in the main text, especially related with the plane group discussed in the paper.

Reply: We thank the referee for this suggestion. We added the general formula to analyze the symmetry properties of the piezoelectric tensor in the main text and added a more detailed analysis for the relevant plane groups in the “Method” section.

(6) There are too many acronyms, which disturb the readability of the manuscript.

Reply: We thank the referee for this comment. We abandoned some acronyms like VCN for valley Chern number and FTP for fine-tuning parameter, and listed all the remaining ones in the “Method” part for reference.

Reply to Reviewer #3

Reviewer #3 (Remarks to the Author):

In this manuscript, Yu and Liu have thoroughly studied the piezoelectric response across topological phase transitions in two dimensional time-reversal invariant systems. Using both analytical arguments and numerical calculations, the authors have made the strong statement that the piezoelectric response would exhibit a jump across the topological transitions in all 2D time-reversal invariant insulators which break inversion symmetry (C_{2z} symmetry). Moreover, the authors have classified the band-touching events for all 2D plane groups breaking inversion symmetry (preserving time-reversal symmetry), which will be very useful for future theoretical research on 2D materials. The authors also illustrate these ideas using specific material examples, i.e., HgTe/CdTe (111) thin films, and BaMnSb₂, which would provide useful guidelines to experimentalists. The calculations for BaMnSb₂ are indeed quite reliable and practical. Therefore, I would strongly recommend this paper for publication on Nature Communications. I have two questions and suggestions as follows:

1) Does Eq.(3) apply to systems that break time-reversal (TR) symmetry? Or does it rely on the existence of a smooth gauge throughout the Brillouin zone? If it is a general

formula which also applies to TR-breaking systems and is completely gauge independent, then it would be interesting to discuss the change of the piezoelectric response across the topological transition in a Chern insulator with broken inversion symmetry.

Reply: We sincerely appreciate the positive and encouraging comments of this referee.

As for the first question, the referee raises an interesting future direction about generalizing the result to the topological quantum phase transitions in Chern insulators. However, this generalization requires caution as discussed in the reply to Question 2 of Referee #1.

2) There are both electronic and ionic contributions to the piezoelectric response. The former is what has been discussed in this manuscript. The ionic contribution may be important in a given insulating state, but the change of the ionic contribution to the piezoelectric response is expected to be small and continuous across the topological transition, and this distinguishes the electronic contributions (to the piezoelectric response) as a unique probe of the topological transition. I think it would be nice if the authors could point this out.

Reply: We sincerely appreciate the suggestion given by the referee. Indeed, according to Ref. [15], Eq. (3) is derived within the clamped-ion approximation where the ions exactly follow homogeneous deformation and thus do not contribute to piezoelectricity. In reality, the ion contribution might be nonzero, but as the referee pointed out, it should vary continuously across the topological phase transition of electronic band structures, indicating the validity of the clamped-ion approximation in our study. We added this discussion in the updated version.

List of Changes

1. Replaced “VCN” and “FTP” in the main text by “valley CN” and “fine-tuning parameter”.
2. Added Ref. [17,18,45,62].

3. In the Introduction, “The two-fold rotation C2 or the 2D inversion restricts the PET to zero in the other 10 PGs (17).” was revised to “The two-fold rotation C2 (with the axis perpendicular to the 2D plane) or the 2D inversion...of the 2D material.”
4. In the subsection “Classification of Direct 2D TQPTs and PET jumps for 7 PGs” of the Results, “In particular, a systematic classification of GC forms for 2D PGs has not been implemented.” was changed to “While the GC between non-degenerate states...typically enough for experimental predictions.” In the next sentence, “systematic” was replaced by “comprehensive”. At the end of this subsection, “Based on these results, we propose the following criteria...the layered material BaMnSb₂, which are studied in the following.” was added.
5. In the subsection “HgTe/CdTe Quantum Well” of the Results, “To be concrete, we fix the width of the QW at $d = 62 \text{ \AA}$.” was revised to “In order to realize the GC at a nonzero value of the electric field, we fix the width of the QW at $d = 62 \text{ \AA}$, away from 65 \AA .”
6. In the subsection “Layered Material BaMnSb₂” of the Results, “Since the tunneling along the z direction... as a quasi-2D material (48).” was changed to “Owing to the insulating Mn-Sb layers, the tunneling...as a quasi-2D material (51).” Near the end of this subsection, “According to Fig. 3(d), both the jump and background are of the same order... the jump is experimentally measurable.” was added.
7. In the Conclusion and Discussion, “Despite the similarity between Eq. (3) and the expression of CN, the generalization ...of polarization (62).” was added to the last paragraph. “We notice that the dynamical piezoelectric...behave in topological semimetals.” was changed to “We notice that although the dynamical piezoelectric... across the transitions between insulating and semimetal phases.”
8. In Methods, the subsections “List of Acronyms”, “Expression for the PET”, and “Valley CN” were added. In subsections “PGs $p1m1$, $c1m1$, and $p1g1$ ”, “PG $p3$ ”, and “PGs $p31m$ and $p3m1$ ”, the sentence “The U symmetry in these three PGs requires...allowed to be nonzero.”, the sentence “Owing to C3, the PET...”, and the sentence “C3 and m_x span the point group...” were revised to contain more details.

9. Scenario numbers were added to Fig. 1. In the caption of Fig. 1, “The figures are first grouped according to the PGs and then ordered ... the names of the PGs.” was added.
10. In Tab. 1, “GC with 1 FTP” was replaced by “Codim-1 GC”. In its caption, “ “Codim-1 GC” means the GC cases with 1 fine-tuning parameter or codimension 1.” was added.
11. More people were added to the Acknowledgement.
12. The supplementary material was revised to contain more details and incorporate the change made in the main text.

Reviewers' comments:

Reviewer #1 (Remarks to the Author):

Thank you for addressing my questions in the revised manuscript. My recommendation is to publish without further revision.

Reviewer #2 (Remarks to the Author):

I checked the response letter and revised manuscript and found that the authors properly answered all my questions and comments, except the one about the valley Chern number. I still think that the change of the valley Chern number introduced by the authors is not physically meaningful. In the case of the Z_2 invariant, for example, if there is a jump of Z_2 invariant across a gap-closing, it means that the Z_2 invariant of one insulator is zero while that of the other insulator is one. Namely, the change of the Z_2 invariant across a band crossing actually describes the change of the bulk topological invariants of two insulators related by the band crossing. However, in the case of the valley Chern number, I am sure that one cannot make such a connection. Namely, even if the k -local effective Hamiltonian predicts a discrete jump of the valley Chern number, it does not mean that the valley Chern number of one insulator is zero while the valley Chern number of the other insulator is one. If the authors disagree to my opinion, I would like to ask the author to show a numerical evidence, supporting their theory, for the discrete jump of valley Chern numbers using tight-binding Hamiltonian. Namely, using a lattice model, the authors should show that the valley Chern numbers of two insulators (computed by a lattice model), connected by a gap-closing (occurring at generic momenta) are indeed different by one. As I pointed out in my previous report, since the gap-closing points are located at a generic point, it is difficult to impose any physical meaning to the valley Chern number in terms of the lattice model. In this case, it is incorrect to state that the valley Chern number changes by an integer value across a band crossing since the valley Chern number is not physically meaningful in view of the lattice model. I think the author should either show numerical evidence using a lattice model disproving my statement, or properly revise the description about the change of the valley Chern number. Since all the other questions or comments from my previous reports are properly answered by the authors, if the issue related with the valley Chern number can be properly treated, I think I can recommend this paper for publication.

Reviewer #3 (Remarks to the Author):

I think the authors have appropriately responded to all the questions and suggestions from the referees, so I recommend the manuscript for publication.

Reviewer #1 (Remarks to the Author):

Thank you for addressing my questions in the revised manuscript. My recommendation is to publish without further revision.

Reply: We thank the referee for positive comments.

Reviewer #2 (Remarks to the Author):

I checked the response letter and revised manuscript and found that the authors properly answered all my questions and comments, except the one about the valley Chern number. I still think that the change of the valley Chern number introduced by the authors is not physically meaningful. In the case of the Z2 invariant, for example, if there is a jump of Z2 invariant across a gap-closing, it means that the Z2 invariant of one insulator is zero while that of the other insulator is one. Namely, the change of the Z2 invariant across a band crossing actually describes the change of the bulk topological invariants of two insulators related by the band crossing. However, in the case of the valley Chern number, I am sure that one cannot make such a connection. Namely, even if the k-local effective Hamiltonian predicts a discrete jump of the valley Chern number, it does not mean that the valley Chern number of one insulator is zero while the valley Chern number of the other insulator is one. If the authors disagree to my opinion, I would like to ask the author to show a numerical evidence, supporting their theory, for the discrete jump of valley Chern numbers using tight-binding Hamiltonian. Namely, using a lattice model, the authors should show that the valley Chern numbers of two insulators (computed by a lattice model), connected by a gap-closing (occurring at generic momenta) are indeed different by one. As I pointed out in my previous report, since the gap-closing points are located at a generic point, it is difficult to impose any physical meaning to the valley Chern number in terms of the lattice model. In this case, it is incorrect to state that the valley Chern number changes by an integer value across a band crossing since the valley Chern number is not physically meaningful in view of the lattice model. I think the author should either show numerical evidence using a lattice model disproving my statement, or properly revise the description about the change of the valley Chern number. Since all the other questions or comments from my previous reports are properly answered by the authors, if the issue related with the valley Chern number can be properly treated, I think I can recommend this paper for publication.

Reply: The only concern of this referee is on the valley Chern number (VCN). The referee proposed two main points: (i) the VCN is not physical in lattice models as shown by the statement “the valley Chern number is not physically meaningful in view of the lattice model”, and (ii) the change of VCN across a gap closing is not physical in lattice models as shown by the statement “the change of the valley Chern number introduced by the authors is not physically meaningful”. As discussed below, the first point does not imply the second, and we agree with the point (i) but disagree with the point (ii).

For the point (i), we would like to emphasize that we have never claimed that the VCN is quantized or physical in any insulating phases. In the VCN section of the Methods, we explicitly stated that “the valley CN on one side of the GC is not necessarily quantized to integers since the integral of Berry curvature is not over a closed manifold.”, which coincides with the referee’s statement that “even if the k -local effective Hamiltonian predicts a discrete jump of the valley Chern number, it does not mean that the valley Chern number of one insulator is zero while the valley Chern number of the other insulator is one.” Therefore, we agree with the point (i) raised by the referee and have included it in the previous version.

However, we disagree with the point (ii) of the referee. We believe the change of VCN across a gap closing is always quantized and has physical meaning when valleys are well-defined, even in lattice models. The quantization of the VCN change has been demonstrated in effective models by patching the two low-energy effective models on the two sides of the gap closing at large momenta, as written in the VCN section of the Methods. To fortify our statement, we build a toy tight-binding model for $p1m1$, which can realize the VCN gap closing case in Fig. 1(d) by tuning one parameter. (See Fig. S4(a) in Supplementary Materials(SM).) The change of valley CN across the gap closing can be illustrated by tracking the integration of Berry curvature over one quarter of 1BZ, and the quantized change is shown in Fig. S4(c) in SM.

One direct physical consequence of the quantized change of VCN is the gapless interface mode in a domain wall structure that consists of two phases separated by the gap closing. This issue has been addressed in bilayer graphene systems (Ref. (46)). Following the logic in Ref. (46), we use the tight-binding model to construct the domain wall structure and choose the interface direction such that the projections of the four valleys are well separated (Fig. S4(d) of SM). As shown in Fig. S4(e) of SM, the quantized change of VCN for each valley is directly reflected by the number of gapless domain-wall modes local to the valley. The gapless domain-wall modes can be observed with transport or optical measurements (for bilayer graphene cases, see Ref. (70) and the references therein), serving as a physical consequence of the quantized change of VCN.

In sum, the change of VCN across a gap closing is quantized and physically meaningful, while the VCN itself is not. We emphasize this point in the updated version of the main text and discuss it in the new section of the SM with more details.

Reviewer #3 (Remarks to the Author):

I think the authors have appropriately responded to all the questions and suggestions from the referees, so I recommend the manuscript for publication.

Reply: We thank the referee for the recommendation for publication.

List of Changes

1. Added Ref. (46) and (70).
2. Added “We would like to emphasize that although ... (See the Methods for more details.)” in the section “Classification of Direct 2D TQPTs and PET jumps for 7 PGs”.
3. Added “(See SM for more details.) “ and “One physical consequence of the quantized change of ... domain-wall mode with a tight-binding model in SM.” in the “Valley CN” section of “Methods”.
4. Added section “VCN in Tight-Binding Model” in SM.

REVIEWERS' COMMENTS:

Reviewer #2 (Remarks to the Author):

I think the authors misunderstood my question about the physical meaning of the valley Chern number. In the revised manuscript, the authors defined the valley Chern number in terms of the Berry curvature integrated over a quarter of the Brillouin zone. My question was that the Berry curvature integrated over a quarter of the Brillouin zone, for instance, cannot be related with any physical quantities such as spin, orbital etc. Because of the same reason, the stability of the domain wall metallic states proposed by the authors is also not guaranteed in general. Although I do not think that the response of the authors about this issue is clearly given, it does not significantly affect the value of the paper. Also the discussion about the valley Chern number in the revised manuscript is self-explanatory, I believe that the readers can properly judge the meaning of the valley Chern number proposed by the authors in a proper context. Given this, I recommend the revised manuscript for publication without further delay.

Reviewer #2 (Remarks to the Author):

I think the authors misunderstood my question about the physical meaning of the valley Chern number. In the revised manuscript, the authors defined the valley Chern number in terms of the Berry curvature integrated over a quarter of the Brillouin zone. My question was that the Berry curvature integrated over a quarter of the Brillouin zone, for instance, cannot be related with any physical quantities such as spin, orbital etc. Because of the same reason, the stability of the domain wall metallic states proposed by the authors is also not guaranteed in general. Although I do not think that the response of the authors about this issue is clearly given, it does not significantly affect the value of the paper. Also the discussion about the valley Chern number in the revised manuscript is self-explanatory, I believe that the readers can properly judge the meaning of the valley Chern number proposed by the authors in a proper context. Given this, I recommend the revised manuscript for publication without further delay.

Reply: We thank the reviewer for the recommendation of the publication.

The reviewer believes that we misunderstood his question about the physical meaning of the valley Chern number (VCN) and our previous answer to his question is not clear. Specifically, the reviewer believes that a clear answer should relate VCN to physical quantities like spin, orbital, etc, while our previous answer only pointed out the gapless domain-wall modes related to the VCN change.

First, we would like to emphasize that in the last review report, the reviewer did not require us to relate VCN to physical quantities like spin, orbital, etc, and instead the reviewer just wanted us to show the physical meaning of the VCN. Second, we believe the physical meaning should *not* be confined to spin, orbital, and so on, and instead it should include any property that can be experimentally observed. In the last reply, we pointed out that the gapless domain-wall mode can serve as the physical consequence of VCN change since it is experimentally observable and stable against *any perturbation that keeps valleys well-defined*. We also notice that the terminology of VCN has been previously used in a number of literatures on 2D materials (to name just a few: Ref. [1-4]). Therefore, we believe that our previous answer is appropriate to the question raised in the last report and clarifies the physical meaning of VCN change.

References:

[1] Zhang, Fan, Allan H. MacDonald, and Eugene J. Mele. "Valley Chern numbers and boundary modes in gapped bilayer graphene." *Proceedings of the National Academy of Sciences* 110.26 (2013): 10546-10551.

[2] Ju, Long, Zhiwen Shi, Nityan Nair, Yinchuan Lv, Chenhao Jin, Jairo Velasco Jr, Claudia Ojeda-Aristizabal et al. "Topological valley transport at bilayer graphene domain walls." *Nature* 520, no. 7549 (2015): 650-655.

[3] Ren, Yafei, Zhenhua Qiao, and Qian Niu. "Topological phases in two-dimensional materials: a review." *Reports on Progress in Physics* 79, no. 6 (2016): 066501.

[4] Khanikaev, Alexander B., and Gennady Shvets. "Two-dimensional topological photonics." *Nature photonics* 11, no. 12 (2017): 763-773.